# Fine-grained Hallucination Detection and Editing for Language Models

**Abhika Mishra**[1], **Akari Asai**[1], **Vidhisha Balachandran**[2], **Yizhong Wang**[1]
**Yulia Tsvetkov**[1], **Graham Neubig**[2], **Hannaneh Hajishirzi**[1,3]
[1]University of Washington
[2]Carnegie Mellon University
[3]Allen Institute for AI
{abhikam,akari,yizhongw, yuliats, hannaneh}@cs.washington.edu,
{vbalacha, gneubig}@cs.cmu.edu

## Abstract

Large language models (LMs) are prone to generate factual errors, which are often called *hallucinations*. In this paper, we introduce a comprehensive taxonomy of hallucinations and argue that hallucinations manifest in diverse forms, each requiring varying degrees of careful assessments to verify factuality. We propose a novel task of **automatic fine-grained hallucination detection** and construct a new evaluation benchmark, FavaBench, that includes about one thousand fine-grained human judgments on three LM outputs across various domains. Our analysis reveals that ChatGPT and Llama2-Chat (70B, 7B) exhibit diverse types of hallucinations in the majority of their outputs in information-seeking scenarios, highlighting the need to build fine-grained systems. To this end, we train Fava, a powerful retrieval-augmented LM by carefully creating synthetic data to detect and correct fine-grained hallucinations. Our automatic and human evaluations show that Fava significantly outperforms retrieval-augmented ChatGPT and GPT-4 on fine-grained hallucination detection. Furthermore, Fava outperforms widely-used hallucination detection systems on binary detection and shows effectiveness in editing to improve the factuality.[1]

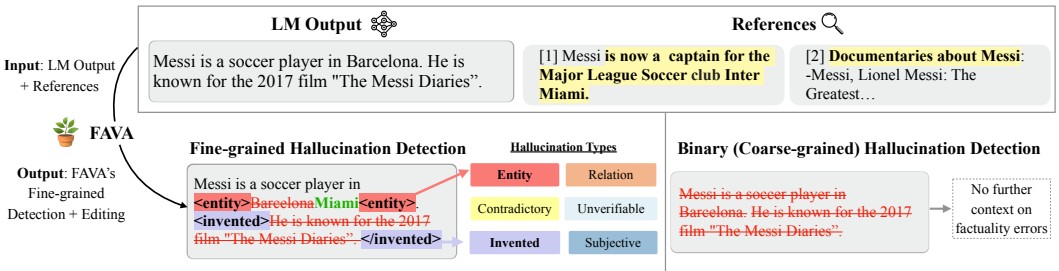

Figure 1: Overview of our taxonomy, fine-grained hallucination detection task, and Fava.

## 1 Introduction

Large language models (LMs; Brown et al. 2020) can generate highly fluent and plausible text. However, these models are prone to produce factually incorrect or unverifiable statements, often called *hallucinations*. This impedes their deployment in real-world applications for information-seeking contexts (Mallen et al., 2022; Asai et al., 2024). Prior work on hallucinations in natural language generation (NLG) often assumes the presence of a specific

---

[1]Our code, data, and demo are available at https://fine-grained-hallucination.github.io/.

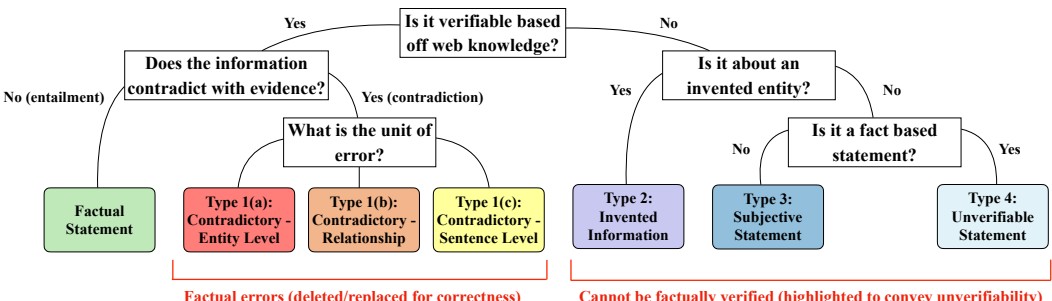

Figure 2: An overview of our fine-grained hallucination taxonomy. We identify 6 fine-grained types representing diverse hallucinations in LM-generated text.

reference text and focuses on studying faithfulness to the references (Ji et al., 2023). On the contrary, escalating apprehensions have been articulated about LM generations that are not grounded in any specific source text, but rather in world knowledge (Zhang et al., 2023).

Several recent work studies automatic hallucination detection (Min et al., 2023) or editing outputs (Gao et al., 2022) to address such LM hallucinations. These systems typically categorize hallucinations into simplistic binary distinctions like *factual* or *not factual* (Figure 1 right) or mainly focus on entity-level errors. We argue that hallucinations manifest in diverse forms, each requiring varying degrees of careful assessments to verify factuality. Entity-level contradictions are usually evident and can be easily rectified with a single reference. Conversely, errors involving fabricated entities (e.g., The Messi Diaries in Figure 1) demand thorough verification across multiple sources. This underscores the need for a more fine-grained approach to detect hallucinations.

In this work, we propose **automatic fine-grained hallucination detection** (Figure 1 left), a new task requiring a system to provide precise identification of hallucination sequences, discern different types based on a taxonomy, and suggest refinements. We focus on hallucinations in information-seeking scenarios, where grounding to world knowledge matters. Our taxonomy (Figure 2) hierarchically classifies hallucinations in LM generations into six categories. Our taxonomy is based on iterative pilot studies with NLP experts, to ensure comprehensive coverage of all significant types of hallucinations.

We construct a new fine-grained hallucination benchmark, FAVABENCH, by carefully annotating approximately 1,000 responses of three widely used LMs (Llama2-Chat 7B, 70B and ChatGPT[2]) to diverse knowledge-intensive queries. Each response is annotated at the span level to identify hallucinations, including erroneous subspace, types, and potential refinements. Our analysis reveals all models include at least one hallucination in the majority of their responses (e.g., 70.2% in Llama2 7B and 59.8% in ChatGPT). Besides the widely-studied entity-level errors, other types like unverifiable sentences make up over 60% of LM-generated hallucinations, highlighting the urgent need for fine-grained detection.

Yet, prior methods often only provide statement-level binary hallucination predictions. To advance fine-grained hallucination detection, we introduce FAVA, a new retrieval-augmented LM that can identify and mark hallucinations at span-level using a unified syntax (Figure 1, left). We design an LM-based synthetic data generation and train FAVA on the 35k resulting instances. We compare FAVA with state-of-the-art LMs on fine-grained hallucination detection based on FAVABENCH. FAVA significantly outperforms ChatGPT with and without external knowledge by 23.7% on fine-grained hallucination detection. On binary hallucination detection, FAVA outperforms a widely-used system, FActScore (Min et al., 2023) in addition to other strong baselines leveraging GPT4 or ChatGPT. FAVA also effectively corrects hallucinations in diverse LM outputs—FAVA editing improves a factuality score (Min et al., 2023) of Alapaca 7, 13B, and ChatGPT by 4.4, 9.3 and 3.3%, respectively.

---

[2]We use `gpt-3.5-turbo-0301` throughout this work.

| Type | Example | Chat | L-7b | L-70b |
|------|---------|------|------|-------|
| Entity | Lionel Andrés Messi was born on June ~~12~~ 24, 1987. | 42.7% | 48.6% | 55.7% |
| Relation | Lionel Messi ~~acquired~~ **was acquired by** PSG. | 4.7% | 3.4% | 2.8% |
| Sentence | **Messi was never captain for Argentina football team.** | 18.9% | 15.7% | 19.9% |
| Invented | **Messi is known for his famous airplane kick.** | 14.2% | 17.4% | 9.5% |
| Subjective | Lionel Messi is **the best soccer player in the world**. | 9.9% | 8.6% | 5.8% |
| Unverifiable | **When free, Messi enjoys singing songs for his family.** | 9.6% | 6.3% | 6.3% |

Table 1: Distribution of different errors across ChatGPT (Chat), Llama2-Chat-7B (L-7B) and Llama2-Chat-70B (L-70B) outputs and examples of hallucinations for each type.

## 2    Related Work

**Hallucinations in NLG.** Prior taxonomies proposed for summarization (Pagnoni et al., 2021), text simplifications (Devaraj et al., 2022) or knowledge-grounded dialogue (Dziri et al., 2022) often assume the existence of specific source text and focus on faithfulness to the source (Ji et al., 2023). As this is qualitatively different from LMs' factual hallucinations grounded in world knowledge, we extend prior work and build a new taxonomy for hallucinations in LM outputs.

**Detecting and editing hallucinations in LMs.** Several recent or concurrent studies propose methods to predict factuality by giving binary labels of a statement being factual or not (Manakul et al., 2023; Min et al., 2023), focusing on entity-level hallucination editing (Gao et al., 2022; Chen et al., 2023), or concentrating purely on detection (Li et al., 2024). Recent surveys have attempted to categorize hallucinations in LM generations, often stretching the definition of hallucinations to broad error types in LM-generated text, which cannot be traced to specific spans in text (Huang et al., 2023; Zhang et al., 2023; Rawte et al., 2023). We propose a fine-grained taxonomy for hallucinations in long-form text generation for information-seeking context to improve factual consistency. While prior work often develops binary factuality verification systems on top of proprietary LMs (Chern et al., 2023), we train a fine-grained detection system using our carefully designed data generation pipeline.

**Fact verification for human-written claims.** While research on hallucination detection focuses on identifying errors in model-generated text, a related area of research focuses on identifying factual inaccuracies in human written claims (Bekoulis et al., 2021). Many large-scale datasets have been proposed for fact verification given Wikipedia (Thorne et al., 2018), scientific articles (Wadden et al., 2020) or news documents (Wang, 2017). Several systems have been developed for fact-checking in these settings (Thorne & Vlachos, 2018; Schuster et al., 2021; Nakov et al., 2021), but have not been tested for LM-generated text. While our work focuses on hallucination detection in LM generations, FAVA can be adapted to fact-check human written claims.

## 3    Fine-grained Hallucination Detection

**Focus and definitions.** We focus on open-ended text generation given information-seeking queries which often require factual knowledge. We define hallucinations as *factual errors or unverified statements given external world knowledge*. We operationalize world knowledge as documents from the web that are most relevant to the given statement/query according to a search algorithm.[3]

### 3.1    Hallucination Taxonomy

Based on our definition of hallucinations, we build a hierarchical taxonomy to categorize them into fine-grained error types. Inspired by prior task-specific taxonomies (Pagnoni et al., 2021; Devaraj et al., 2022), we introduce new categories to describe more complex

---

[3]Errors in common sense, numerical, or logical reasoning are out of the scope of this work.

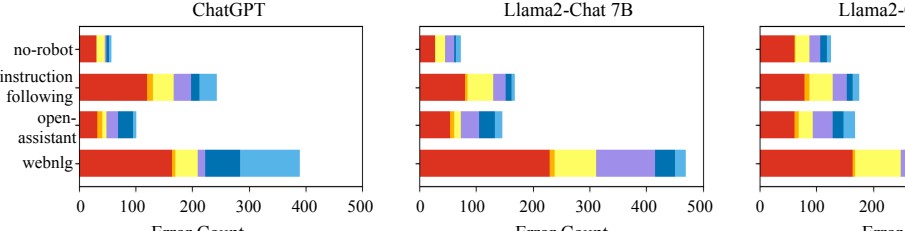

Figure 3: Distribution of hallucination types in ChatGPT, Llama2-Chat-7B and Llama2-Chat-70B outputs across four datasets of diverse information-seeking queries.

errors surfacing in LM generations. We conducted a pilot annotation with nine NLP experts to ensure good coverage across diverse error types. Figure 2 shows our taxonomies to classify LM hallucinations. Table 1 shows examples of each type of hallucination. We group hallucinations into two major categories: statements that contradict world knowledge (Type 1) and unverifiable statements (Types 2, 3, and 4).

**(1a)** `Entity` : Contradictory entity errors are a sub-category within Type 1, where an entity in a statement is incorrect and changing that single entity can make the entire sentence factually correct.

**(1b)** `Relation` : Contradictory relation errors are another sub-category within contradictory statements where a semantic relationship (e.g., verbs, prepositions, or adjectives) in a statement is incorrect.

**(1c)** `Sentence` : Contradictory sentence errors refer to cases where a full statement entirely contradicts relevant evidence from the web, and cannot be solved via phrase-level edits.

**(2)** `Invented` : Invented errors refer to statements where the LM generates an entirely fabricated entity that doesn't exist based on world knowledge. Fictional entities in creative work aren't included.

**(3)** `Subjective` : Subjective errors refer to expressions about existing entities that lack universal validity. These statements often do not contain facts and are influenced by personal beliefs or opinions.

**(4)** `Unverifiable` : These are statements where the LM output contains facts, but no retrieved evidence from the web can directly support or contradict the fact (e.g., private details). [4] [5] While `Entity` or `Relation` are often phrase-level and can be fixed by minimal editing erroneous phrases, other error types can be an entire sentence or part of a sentence and should be removed from a response to make it precisely factual.

### 3.2 Tasks and Metrics

We introduce the two tasks of identifying and editing fine-grained factual errors in LM outputs. Given an input query $x$ and a corresponding LM output $y$, our tasks require systems to identify all of the factual errors in $y$. Each error $e$ consists of $(e^{text}, e^{type})$, indicating the factually incorrect text spans and their error types among our taxonomies, respectively. We evaluate systems' abilities concerning identifying fine-grained error types in the model-generated text $e^{type}$ (**Task 1**; Figure 1 < `Entity` >) and editing factual errors $e^{text}$ accordingly (**Task 2**; Figure 1 Barcelona → Miami).

**Task 1: Fine-grained hallucination detection.** In this task, the system is expected to identify fine-grained errors in an LM output. Due to the subjectivity of span-level annotations, for automatic evaluation, we evaluate systems' abilities to detect whether an error type $t$ exists in a sentence $s_i \in y$. Given an output $y$ consisting of $L$ sentences, we assume the availability of ground-truth error type annotations $e_i^{*t} \in \{\texttt{TRUE}, \texttt{FALSE}\}$, which is a binary label of an error type $t$ existing in the $i$th sentence (`TRUE`) or not (`FALSE`). Following the fact verification

---

[4]Differing from `Subjective` errors which focus on opinionated statements lacking facts, `Unverifiable` specifically focuses on fact-based statements that just lack evidence.

[5]While `Invented` and `Unverifiable` are highly related, `Invented` are the hallucinations where we can verify that some core entities or subjects of the sentences don't exist.

literature (Thorne et al., 2018; Schuster et al., 2021; Feng et al., 2023) of computing precision and recall, we measure those metrics while we extend them for each error type. For each type, a system predicts $e_i^t$ and we evaluate precision and recall as:

$$\text{Prec}^t = \frac{\sum_{i \in L} \mathbb{1}[e_i^t = e_i^{*t}]}{\sum_{i \in L} \mathbb{1}[e_i^t = \text{TRUE}]}, \quad \text{Recall}^t = \frac{\sum_{i \in L} \mathbb{1}[e_i^t = e_i^{*t}]}{\sum_{i \in L} \mathbb{1}[e_i^{*t} = \text{TRUE}]} \quad (1)$$

Precision indicates the proportion of how many of the model's predictions of an error type $t$ existing in the $i$th sentence is correct, while recall indicates how many of the error sentence TRUE is identified by the model. For final score, we compute the F1 scores averaged over six error types as follows:

$$\frac{1}{|\mathcal{E}|} \sum_{t \in \mathcal{E}} \frac{2 \times \text{Prec}^t \times \text{Recall}^t}{\text{Prec}^t + \text{Recall}^t} \quad (2)$$

Fine-grained detection can be simplified into a binary classification task, in which a system predicts if a sentence $s_i$ includes any factual errors or not.[6]

**Task 2: Hallucination editing.** Some hallucinations can be fixed with minimal span-level edits, while others need to be removed or marked as unverifiable. In the editing task, the system is expected to suggest fine-grained edits to improve the factuality of the LM output. We evaluate a system's ability to improve the factuality of given output $y$, by comparing scores estimated by off-the-shelf systems, such as FActScore (Min et al., 2023), as: $f(\hat{y}) - f(y)$, where $f$ indicates an estimated factuality scores and $\hat{y}$ indicates edited output.

# 4 Benchmark: FAVABENCH

FAVABENCH consists of around 1k fine-grained annotations on three LM responses to queries in multiple domains.

**Source prompts.** The source prompts include a collection of 200 information-seeking queries, spanning four different data sources. See examples in Appendix Table 6.

- 50 Knowledge-intensive queries from Open Assistant (Köpf et al., 2023), by prompting GPT-4 to judge whether each query from the dataset requires world knowledge.
- 50 Open QA prompts from the No Robots dataset (Rajani et al., 2023).
- 50 instructions requiring more reasoning and knowledge by the authors.
- 50 synthetically-created prompts that require more fine-grained knowledge, by converting data-to-text WebNLG dataset (Gardent et al., 2017) using a template.

**Annotation details and quality.** We obtain responses to the collected source prompts using ChatGPT (`gpt-3.5-turbo-0301`; Ouyang et al. 2022) Llama2-Chat 7B and Llama2-Chat 70B (Touvron et al., 2023) in a zero-shot manner and collect 600 responses to our diverse information-seeking prompts. We recruited 20 students (ten undergraduate and ten NLP graduate students) to annotate the factual accuracy of the responses based on our proposed taxonomy. Each instance is annotated by two annotators who completed 45-minute in-person and virtual training sessions.[7] We provide annotators with detailed instructions, examples, and training to ensure high-quality annotations. Our annotation interface and details are in Appendix A and examples in Appendix Table 8. To validate our annotation quality, we calculated inter-annotator agreement using Cohen kappa scores. Our annotators have high agreement in detection across passages, with 75.1% agreement in detection at the sentence level and 60.3% agreement in exact error type detection.

**Analysis on annotated data.** Figure 3 presents a detailed breakdown of distributions across fine-grained categories in the three domains. 59.8%, 70.2% and 64.9% of the responses of ChatGPT, Llama2-Chat 7B and Llama2-Chat 70B include at least one hallucination, respectively. Entity is the most widely recognized error type, making up 48.3% of the detected

---

[6]This is similar to FActScore (Min et al., 2023), while we predict and evaluate factuality at the sentence level without extracting atomic facts. We compare FAVA against FActScore on this task.

[7]Annotating each response takes approximately 10 minutes, and we pay USD 3.5 for annotation.

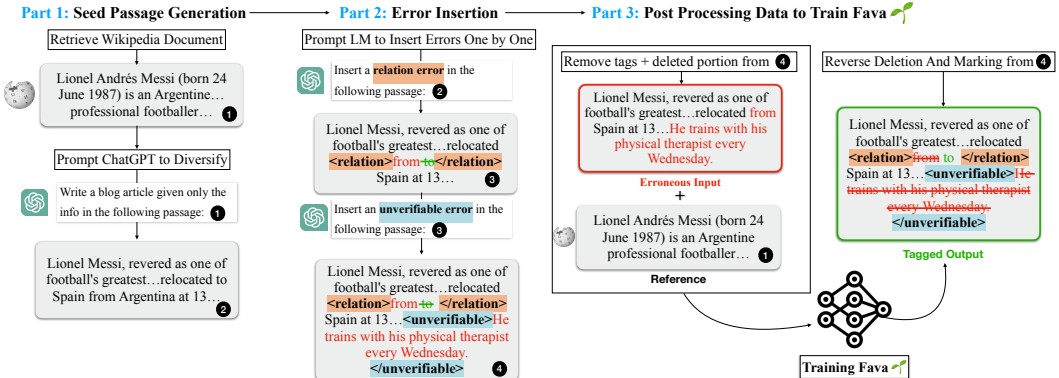

Figure 4: Overview of high-quality synthetic data generation process in FAVA. FAVA leverages powerful instruction-tuned models to carefully insert errors into factually accurate statements and produces diverse error types based on our proposed taxonomy.

errors. Yet, there are diverse types of errors prevalent like invented statements or contradictory statements, which make up 14.1% and 18.1% of the detected errors, respectively. The error distributions vary across different source prompts and LMs. Invented are more common in Llama2-Chat models. Fewer errors in No Robots subsets than other subsets may be because their seed prompts are less knowledge-intensive, or ask about popular factual knowledge (e.g., *How long was the Revolutionary War?*), which is often memorized by LMs (Mallen et al., 2022).

# 5   Model: FAVA

We introduce FAVA (**FA**ct **V**ericaton with **A**ugmentation), a model for fine-grained hallucinations detections and editing. FAVA is trained on high-quality synthetic training data to identify hallucinations, incorporating retrieved knowledge. FAVA consists of two components: a retriever $\mathcal{M}_{ret}$ and an editing LM $\mathcal{M}_{edit}$. $\mathcal{M}_{ret}$ takes the original output LM $y$ and optionally input prompt $x$ (if applicable), and retrieves top relevant documents: $\mathbf{C} = \mathcal{M}_{ret}(x, y)$. Subsequently, the editing model detects and, if possible, edits factual errors in $y$ given the retrieved context: $\hat{y} = \mathcal{M}_{edit}(x, y, \mathbf{C})$. $\hat{y}$ is an augmented output $y$ interleaved by the error edits with hallucination types as shown in Figure 1. While $\mathcal{M}_{edit}$ can be any LM, in our preliminary experiments, we find that making a state-of-the-art proprietary LM such as ChatGPT to perform fine-grained editing via prompting only is challenging (Table 2). Reliance on such models also hurts reproducibility. Therefore, we generate high-quality synthetic training data with minimal human efforts (Section 5.1) and fine-tune a smaller yet powerful LM on the generated data (Section 5.2).

## 5.1   Synthetic Training Data Curation

To train our $\mathcal{M}_{edit}$ model, we require a large number of training instances, each of which includes $(c, y, y^*)$, where $c$ is the gold context, $y$ is an erroneous LM output, and $y^*$ is an output with error tags and correct editing based on $c$. FAVA learns to take $(c, y)$ and generate $y^*$. Inspired by prior work that leverages LMs to generate synthetic training data (Balachandran et al., 2022; Wang et al., 2023b; Asai et al., 2023), we introduce a new data creation method tailored to our fine-grained hallucination taxonomies. Specifically, our data creation pipeline (refer Figure 4) consists of three steps: **Step 1: seed passage generations** that generates a large, diverse collection of seed passages $c$, **Step 2: error insertions** which inserts factually incorrect statements to generate erronous passages $y$, and **Step 3: post-processing** which curates the final training examples.

**Seed passage generation.** The goal in this step is to collect and generate a large, diverse range of seed passages to ensure FAVA can handle a wide range of text types while maintaining control over factuality. Later on, errors are inserted into these seed passages to generate erroneous text, $y$. We base our seed passage collection on Wikipedia articles and sampled

5k QA pairs from Natural Questions (Kwiatkowski et al., 2019). We paraphrase Wikipedia passages into different genres to enable FAVA to be adaptable and effective across various textual formats. Specifically, we randomly sample 35,074 Wikipedia articles $c$ and use them as our gold reference passages. We then randomly sample one of the pre-specified genre types (e.g. blog article, tweet, etc.) for each passage and use ChatGPT to paraphrase the original passage into the given genre.[8]

**Error insertion.** The goal of this step is to simulate erroneous LM generation $y$, which are passages with factual errors. To control error distributions in the training data and incorporate our hallucination taxonomy, we prompt LMs to insert errors from our taxonomy one by one. We use ChatGPT and GPT-4 interchangeably for six different types.[9] See detailed discussions in Appendix B. Given an instruction and few-shot demonstrations of an error type $e^{type}$, GPT-4 or ChatGPT inserts new errors while retaining previously inserted errors. Models mark phrases or sentences for deletion along with their error type and insert phrases and sentences with insertion tags. Now we have an erroneous text $y$, which inserts multiple errors into the original text $t$.

**Post-processing.** We then post-process data, by swapping and removing the error tags and edits to form a clean erroneous text $y$ as the input and use the original correct passage $y^*$ as the output with edits, for training $\mathcal{M}_{edit}$. We filter out the examples violating our editing rules (e.g., inserting errors without marking them up with tags and error types). We also retrieve additional four relevant Wikipedia paragraphs using Contriever-MSMARCO (Izacard et al., 2022), and randomly mix the order of the references, forming the final references **C**.

**Statistics of data and human evaluation.** After analyzing ablations with varying training instance sizes (Section 7.2), we generated 35,074 training instances, [10] 30,0074 based on Wikipedia passages, and 5,000 based on QA pairs. All of the error types are almost equally distributed ($\sim$15 % each) with 3.1 errors inserted for Wikipedia subsets, and 1.4 errors inserted for the QA pairs on average. We present the error distributions in Appendix Table 10. We conduct human evaluations on randomly sampled 50 examples and find that the generated data meets the expected criterion in most cases. See details in Appendix C.4.

### 5.2 Training and Inference

We train a smaller LM on the generated large-scale training data. In particular, given a training instance $(\mathbf{C}, y, y^*)$, our model $\mathcal{M}_{edit}$ takes $(\mathbf{C}, y)$ as input and learns to predict the edited outputs with tags to represent error type $y^*$ using the standard language modeling objective. We use Llama2-Chat 7B to initialize $\mathcal{M}_{edit}$. We empirically found initializing FAVA with this model gives slightly better performance than the pre-trained Llama2. At inference time, we retrieve the top five documents from Wikipedia,[11] using Contriever-MSMARCO, and input them together with an LM output that may include hallucinations. The model identifies hallucinations, marks phrases or sentences for deletion, and suggests edits.

## 6 Experiments

### 6.1 Experiments for Hallucination Detection

**Evaluation data.** We use our new benchmark, consisting of 902 annotated passages,[12] as our test data. We measure the models' detection performance based on sentence-level per-category classification task as formulated in Section 3.2. We evaluate both **fine-grained detection** and **binary detection** settings. Note that fine-grained hallucination detection is our new proposed task, as prior work mostly focuses on the binary detection setting.

---

[8]See the full list of the genres, instructions, and final ChatGPT-generated text in Appendix Table 9.

[9]We use ChatGPT (gpt-3.5-turbo-0301) for the Type **1** and Type **3** while using GPT-4 for other types.

[10]While we see performance improvements up to 30k, the API costs for training data generation were already $3k, reaching our initial budget.

[11]We use English Wikipedia data from January 2023 and generate embeddings using the Contriever.

[12]We collected 1010 annotated passages including our experimental batches, excluding earlier batches for evaluations.

| | Generator: ChatGPT | | | | | | | Generator: Llama2-Chat 70B | | | | | | |
|---|---|---|---|---|---|---|---|---|---|---|---|---|---|---|
| Editor | ent | rel | con | inv | subj | unv | Avg. | ent | rel | con | inv | subj | unv | Avg. |
| ChatGPT | 19.5 | **28.6** | 40.0 | 11.8 | 7.7 | 0.0 | 18.8 | 24.7 | 15.6 | 26.7 | 11.0 | 17.6 | 12.8 | 24.1 |
| Rt+ChatGPT | 28.1 | 19.2 | 25.5 | 5.4 | 37.7 | 15.5 | 24.4 | 33.7 | 24.2 | 24.0 | 22.2 | 17.8 | 4.7 | 27.8 |
| GPT4 | 38.6 | 16.6 | 17.9 | 22.2 | 50.0 | 17.2 | 34.2 | 55.5 | **60.0** | 21.2 | 15.4 | 2.0 | 25.0 | 42.5 |
| FAVA (ours) | **54.5** | 25.0 | **66.7** | **16.7** | **70.5** | **35.3** | **48.1** | **57.3** | 34.5 | **27.7** | **52.2** | **31.3** | **43.4** | **47.2** |

Table 2: Results on fine-grained detection (metric: F1). Full results are in Appendix.

**Baselines.** Fine-grained hallucination is a new task and we test three baselines that use state-of-the-art proprietary LMs: **ChatGPT** prompts ChatGPT (`gpt-3.5-turbo-0301`) with a carefully designed prompt describing all six categories with two demonstrations. **GPT4** (`gpt-4`) does the exact same but prompts GPT4 instead. We were unable to include GPT4 with retrieval in our baseline due to high costs. Since FAVA uses retrieval augmentation, for comparability we include **Rt+ChatGPT**. This baseline prompts ChatGPT using the same prompt and demonstrations but also includes the top five retrieved documents by Contriever at test time to augment the original prompt.[13] We additionally evaluate **FActScore** (Min et al., 2023) for the binary detection task only. This model-based metric prompts ChatGPT and GPT 3 (`davinci-003`) to decompose a response into a set of atomic facts and verify factuality for each using passages from a designated Wikipedia article.

## 6.2 Experiments for Hallucination Editing

**Evaluation data.** For editing evaluations, we use open-ended text-generation data and evaluate models' factuality based on a metric designed to measure the factuality for the target task. In particular, we use the biography generation task proposed by FActScore (Min et al., 2023) to evaluate the effectiveness of editing to reduce hallucinations based on edited outputs' FActScore results.

**Baselines.** For editing experiments, we use **ChatGPT** and **Rt-ChatGPT**, as well as Llama2-Chat 13B with and without retrieval (**Llama** and **Rt-Llama**, respectively), and prompt them with carefully designed prompts and two demonstrations. For all baselines, we retrieve five documents consisting of one document retrieved by string matching based on the entity name and four documents retrieved by Contriever, which are reranked at test time, using a small cross-encoder.[14]

## 6.3 Human Evaluations

Our automatic evaluations may not fully capture the models' abilities to detect and refine hallucinations, due to the potential subjectivity of our annotations or the performance of the factuality evaluation metric. We evaluate randomly sampled 50 outputs from FAVA as well as the baseline with the highest automatic evaluation score, namely Rt-ChatGPT. We ask human annotators to verify how many of the detection and edits are indeed correct based on the provided retrieved documents. This is similar to our automatic precision evaluation, but instead of coarsely evaluating detection performance at the sentence level, we evaluate the model performance at each detection level.

# 7 Results and Analysis

## 7.1 Results

**Fing-grained detection results.** Table 2 shows the fine-grained detection accuracy, namely F1 scores, and binary prediction F1 of FAVA and baselines. We provide the full precision and recall scores in Appendix Tables 13 and 14, respectively. FAVA significantly outperforms ChatGPT, Ret-ChatGPT, or GPT-4 on both fine-grained error detection and binary error

---

[13]While we also tested strong white box LMs including Llama2-Chat 13B, we found their predictions often show confusion among different categories or violate formats.

[14]https://huggingface.co/cross-encoder/ms-marco-MiniLM-L-6-v2

| | Generator | |
|---|---|---|
| Model | Chat | Llama |
| Chat | 50.1 | 68.4 |
| R+Chat | 64.8 | 72.8 |
| GPT4 | 60.8 | 74.2 |
| FAct | 67.7 | 71.2 |
| FAVA | **79.6** | **80.3** |

Table 3: Results of binary detection.

| | Generator | | |
|---|---|---|---|
| Editor | CGPT | Al-13B | Al-7B |
| No Edit | 66.7 | 42.5 | 38.8 |
| CGPT | 58.6 -8.1 | 42.0 -0.5 | 37.9 -0.9 |
| Rt-CGPT | 62.7 -4.0 | 43.9 +1.4 | 39.2 +0.4 |
| Llama | 52.6 -14.1 | 22.7 -19.8 | 18.6 -20.2 |
| Rt-Llama | 58.7 -8.0 | 48.6 +6.1 | 32.2 -6.6 |
| FAVA | **70.0 +3.3** | **51.8 +9.3** | **43.2 +4.4** |

Table 4: Results of editing.

| Settings | Score |
|---|---|
| No Edit | 42.5 |
| (Top 1) | 44.2 +1.7 |
| (Top 5) | 47.0 +4.5 |
| Reranked | 47.7 +5.2 |
| Entity | 50.1 +7.6 |

Table 5: Ablations of retrieval for editing.

detection. FAVA shows high accuracy on error types such as Sentence , Subjective , Entity . On the other hand, its performance on Invented and Unverifiable is still limited. Those two error types often require intensive search over many documents beyond the top few ones, while FAVA by default only considers the top five documents. Table 3 shows the binary hallucination detection performance of FAVA, baselines models, and FActScore (FAct). FAVA outperforms all other models in detecting hallucinations by a large margin.

**Editing results.** Table 4 shows the results of the editing task on the biography generation task. Our FAVA significantly outperforms prompted ChatGPT or Llama2-Chat 70B, despite being a much smaller model in size. We found that ChatGPT is often confused with different errors and tends to over-predict contradictory or subjective statements. FAVA shows the largest gains in FActScore, showing the effectiveness of editing and error type detection.

**Human evaluation results.** Our human evaluation results are shown in Appendix Table 16. FAVA shows significantly better performance than Rt+ChatGPT on both editing and detection scoring 46.3% and 31.8% more than Rt+ChatGPT on each task respectively. Furthermore, FAVA recognizes more errors, 2.4 errors per passage, than retrieval-augmented ChatGPT, 1.9 errors per passage. These results further demonstrate the strong capabilities of FAVA detecting factual errors in LM outputs.

## 7.2 Analysis

**Effects of data scale.** We assess our model on varying sizes of synthetic training data (10k, 20k, and 30k instances) and examine their performance on fine-grained hallucination detection. Appendix Figure 7 demonstrates that FAVA variants with a larger number of training instances perform significantly better at detecting fine-grained errors.

**Effect of retrieval.** We assess FAVA's editing performance on Alpaca 13B outputs with varying retrieved evidence: top 1 document; top 5 documents; reordering the top 5 documents; top 4 documents and mix them with an introductory paragraph of the target entity (entity matching). Table 5 shows the experimental results. More detailed results are in Appendix Table 18. We found that retrieving the top five documents significantly enhances performance compared to retrieval of only the top one document by 4.5%. Ordering passages using a reranking model gives some improvements, which indicates the non-negligible effect of context order, as reported by prior work (Liu et al., 2023). Including the reference based on entity matching gives a notable enhancement to 50.1%, marking a 3.1% gain from the top five retrievals and 7.6% improvement over the unedited generations. These results indicate although FAVA demonstrates strong capabilities, there's room for improvement in retrieving and incorporating many references.

## 8 Conclusions

We introduce a new task of automatic hallucination detection, built upon our newly introduced taxonomy, hierarchically classifying hallucinations in LMs into six categories. We collect the first human-annotated fine-grained hallucination benchmark, and introduce a new retrieval-augmented LM for fine-grained error editing data following our taxonomies.

Empirical results show that FAVA significantly outperforms strong baselines by a large margin on both editing and fine-grained detection tasks, while still large room for improvements for automated fine-grained error detection and editing.

## Acknowledgement

We thank Sewon Min for fruitful discussions in the early stages and UW NLP members and annonymous reviewers for their insightful feedback on the draft. We thank Stability AI for providing computing to train and evaluate the LMs in this work, and Microsoft Accelerate Foundation Models Research Program for the access to Azure and OpenAI APIs. We thank our annotators who provide high-quality annotations for this work. This work was funded in part by the DARPA MCS program through NIWC Pacific (N66001-19-2-4031), NSF IIS-2044660, and gifts from AI2. We gratefully acknowledge support from NSF CAREER Grant No. IIS2142739, NSF Grants No. IIS2125201, IIS2203097, and gift funding from Google, MSR, and OpenAI.

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

# Appendix

## A  Details of Human Annotations

**Seed query selections.**  Table 6 shows examples of the seed queries for our initial dataset creation.

| Dataset | Example |
|---|---|
| WebNLG | Explain A.C. Cesena, including information about ground, league. |
| Instruction-following | Explain the differences between New York cheesecakes and Basque cheesecakes in detail. |
| Open-Assistant | Can you tell me how tall the burj khalifa is? |
| No Robots | When was Samsung founded? |

Table 6: Examples of source prompts.

**Annotation interface.**  Figure 5 shows the interface of our annotations.

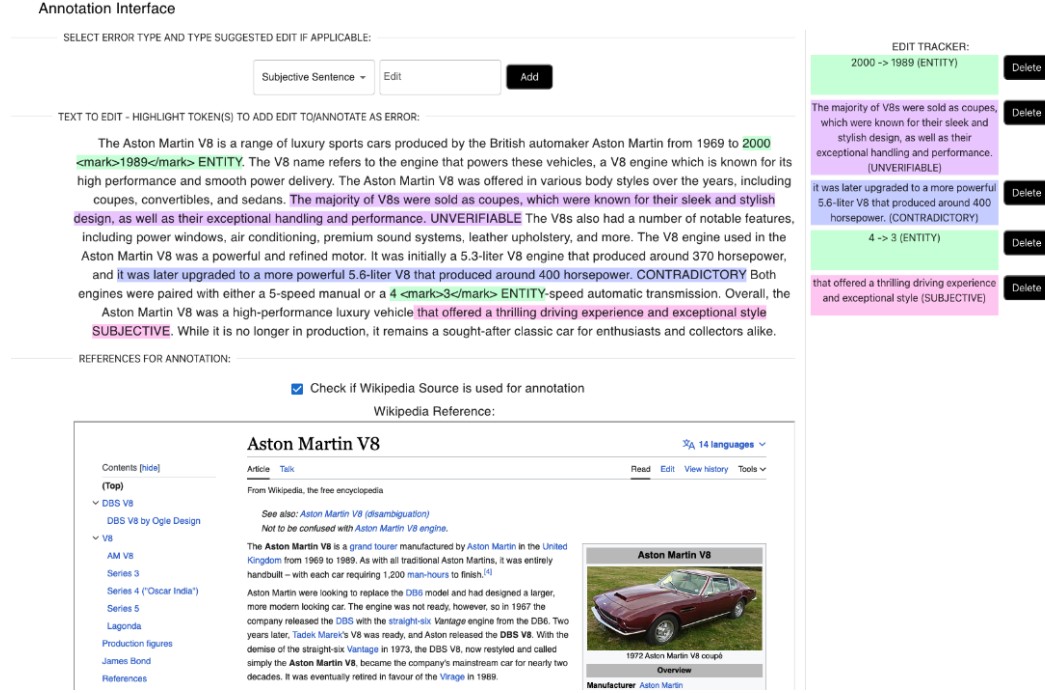

Figure 5: Annotation interface.

**Details of annotations.**  We hired 10 undergrads studying computer science to complete annotations. Each annotator was assigned 60 passages to annotate and had to undergo a 45 minute one on one training session to understand the task and how to navigate the annotation platform. The training session covered an in depth explanation of the six different hallucination types in our taxonomy, included a walk through of how to annotate a passage, and allowed time for annotators to ask any questions.

Annotators were given an instruction document outlining the task, details on the hallucination type, details on how to navigate the annotation platform, and payment details which they looked over during their training session before starting annotation work. Figure 6 shows the instructions on using the annotation platform and payment details provided to annotators.

Instructions

1. Read through the passage provided and **scan for errors or hallucinations** present. You can **use the references provided** below to help with this.
2. For any error or hallucination you identify. **Select the hallucination/error type** you think corresponds to the text you detected as erroneous **from the drop down menu**.

   a. For **entity** and **relational** errors, **type in a proposed edit** to the erroneous text you identified to make the statement factually correct.

3. **Highlight the text** in the passage you want **to annotate**.
4. Click the **add** button to insert your annotation.
5. If you use the wikipedia reference provided, please select the checkbox above the article.

   a. If you use the top 10 google search articles or any other articles from the first page of google, please copy and paste the urls used in the textbox provided at the bottom.
   b. For any unverifiable error types, please add the top 10 urls you looked through to determine it was unverifiable.

6. Click the **next** button to submit your annotation and move on to the next one.
Here is a link to a detailed walkthrough of the platform.

## Payment Details

You will be assigned 60 annotations with the expectation that each annotation takes 10 minutes. For the 600 minutes or 10 hours that you give towards annotations, we will compensate you $200 ($20/ hr)

Figure 6: Annotation instructions and payment details provided to annotators.

**Examples of annotated results.**   Table 7 shows the fine grained hallucination distribution across our annotations. Table 8 shows list of annotated examples from each of our annotation datasets - WebNLG, OpenAssistant, and Instruction Following dataset.

| Type | Percentage |
|---|---|
| Entity Errors | 28.4% |
| Relation Errors | 2% |
| Contradictory Statements | 6.1% |
| Invented Statements | 3% |
| Subjective Statements | 20.2% |
| Unverifiable Statements | 40.3% |

Table 7: Hallucination percentages in our human-annotated data.

| Dataset | Annotated Passage Example |
|---------|---------------------------|
| WebNLG | A Long Long Way is a novel written by Sebastian Barry. It was first published in 2005 by <**entity**>~~Faber and Faber~~ viking press</**entity**>, a <**entity**> ~~British~~ American< **/entity**> publishing company. The novel tells the story of Willie Dunne, an Irish soldier who has a strong sense of loyalty to his country, but also feels torn between his Irishness and his loyalty to the British army during World War I. The novel was critically acclaimed and received several literary awards, <**contradictory**> ~~including the 2005 Dublin Literary Award~~ </**contradictory**> and was shortlisted for the Man Booker Prize in the same year. |
| OpenAssistant | <**subjective**> **The most luxurious airline flying within the continental United States is generally regarded as JetBlue Airways**</**subjective**>, particularly for its Mint business class service. Mint offers a private suite experience, featuring fully lie-flat seats, ample legroom and storage, premium food and beverage options, access to onboard WiFi and live TV streaming, and turn-down service complete with duvets and pillows. |
| Instruction Following | Some of Afghanistan's secondary cricket home grounds include:
1.<**contradictory**>~~Kandahar Cricket Stadium, Kandahar~~</**contradictory**>
2.<**invented**> **Mis Ainak Cricket Ground, Logar**< **/invented**>
3.<**invented**> **Bost University Cricket Ground, Helmand**< **/invented**> |

Table 8: Examples from our human annotations for each dataset.

## B    More Details of Synthetic Data Creation and Training

**Quality of error insertions.**    In our pilot studies, we found that asking ChatGPT to insert multiple error types at the same time easily makes the model misunderstand or get confused with different error types (e.g., swapping or incorrectly identifying an error type).   We also found while ChatGPT is capable of generating more simple types of errors such as Entity , it struggles with generating plausible and difficult perturbations for more nuanced types.

**Text diversification results.**    Table 9 shows the diversified text results.

| Seed Passage Diversification Examples | | |
|---|---|---|
| Genre | Topic | Output |
| Article | Community Reinvestment Act | The Community Reinvestment Act (CRA) is a federal law that encourages commercial banks and savings associations to help meet the needs of borrowers in low- and moderate-income neighborhoods. The Act was passed in 1977 by Congress to reduce discriminatory credit practices against low-income neighborhoods. |
| Essay | National Dodgeball League | The National Dodgeball League is the only professional dodgeball league in the US, and was founded in 2004. It is headed by Commissioner Edward Prentiss out of Hopkins, MN and is composed of 24 professional teams that are divided into the National and American Dodgeball Conferences. In the past, the tournament has featured teams from Canada, Denmark, UK, Japan, New Zealand and Australia. |
| Blog Post | Red Channels | In the beginning of the 1950s, the "Counterattack" journal published a document, "Red Channels," which named 151 individuals in the entertainment industry who were believed to be under Communist manipulation or association. Many of these individuals were being denied employment or had their careers negatively impacted due to their political beliefs or history. "Red Channels" effectively created a blacklist of those named in the document, further limiting their opportunities in the industry. |
| Social Media Post | Sandman | Check out The Sandman - a semi-retired American professional wrestler! He is best known for his career with Extreme Championship Wrestling (ECW), where he held the ECW World Heavyweight Championship a record five times. He also had stints in World Championship Wrestling (WCW), Total Nonstop Action Wrestling (TNA), and World Wrestling Entertainment (WWE). #Wrestling #ECW #WCW #TNA #WWE |
| News Report | I Don't Mind (James Brown song) | James Brown's "I Don't Mind" has made its way up the R&B "Billboard" charts, hitting number four. The track was released in 1961 and also reached number 47 on the Pop "Billboard" charts. Fans will also be able to hear the song on Brown and the Famous Flames album Live at the Apollo. |
| Summary | Robert Lui | Robert Lui was born on February 23, 1990, and is a professional rugby league footballer from Australia. He plays either as a halfback or five-eighth for the Townsville Blackhawks in the Queensland Cup. |
| Speech | George Sperling | Dear fellow Americans, I am here to propose a solution to improve American Sign Language communication. George Sperling suggests that with a sevenfold reduction in the bandwidth for video transmission, we can achieve this. He even argued that the telephone was originally created for the hearing impaired but it became popularized by the hearing community. Let us not forget our roots and make a change for the better. |
| Presentation Intro | Blissful Ignorance Effect | Have you ever wondered why sometimes people who know less about a product seem to enjoy it more than those who have researched it thoroughly? This phenomenon is called the Blissful Ignorance Effect and it's a fascinating topic in consumer behavior studies. Our presentation today will explore this effect and why it happens. |
| Brochure | Nidulariaceae Fungi | Explore the fascinating world of Nidulariaceae fungi! This family, found in most ecological regions, includes five different genera, each with its own unique characteristics. With their tiny egg-filled structures, these fungi are a wonder to behold! |
| Text Message | Ashes and Diamonds | Just found out about this book called Ashes and Diamonds by Jerzy Andrzejewski. It's set during the last few days of WWII. The main character, Maciek, has to kill a Communist soldier. Sounds intense! |

Table 9: Text diversification prompts. Instructions for diversification follow the following format: *"Given a passage, create a(n) [genre] of 3-6 sentences using only the information present in the passage. Do not include any new information not presented in the passage. Passage: [sampled wikipedia paragraph]"*

**Error distribution.** Table 10 shows the distribution of error types across all our generated passages.

| | Percentage |
|---|---|
| Entity | 21.2% |
| Relation | 19.9% |
| Sentence | 15.3% |
| Invented | 14.6% |
| Subjective | 14.1% |
| Unverifiable | 14.9% |

Table 10: Statistics of generated training data

**Training Details.** Our base model is Llama2-Chat 7b trained using 4xA40 GPUs. Our training code is based off Open-Instruct (Wang et al., 2023a)[15]. Table 11 shows the training hyperparameters.

| Precision | Epochs | Weight Decay | Warmup Ratio | Learning Rate | Max. Seq. Length | Batch Size |
|---|---|---|---|---|---|---|
| BFloat16 | 2 | 0 | 0.03 | 2e-5 | 2048 | 128 |

Table 11: Training hyperparameters.

---

[15]https://github.com/allenai/open-instruct/blob/main/scripts/finetune_with_accelerate.sh

# C More Results and Analysis

## C.1 Detection Results

**Llama2-Chat 7B F1 results**   Table 12 reports the detection results for the Llama2-Chat 7B generations on our curated datasets.

| Editor | ent | rel | con | inv | subj | unv | OA | Bi |
|---|---|---|---|---|---|---|---|---|
| | | | Generator: Llama2-Chat 7B | | | | | |
| ChatGPT | 25.2 | 12.6 | 16.1 | 15.0 | 12.4 | 12.8 | 26.5 | 63.4 |
| Rt+ChatGPT | 35.5 | 17.5 | 13.2 | 22.2 | 10.9 | 14.6 | 32.9 | 70.4 |
| GPT4 | 45.4 | 23.1 | 28.8 | 15.4 | 3.4 | 28.6 | 47.3 | 72.1 |
| Fava (ours) | 58.3 | 38.7 | 24.2 | 58.9 | 31.25 | 44.4 | 39.6 | 79.9 |

Table 12: Fine-grained detection F1. OA and Bi indicates overall and binary predictions.

**Overall precision and recall.**   Tables 13 and 14 report the precision and recall on our curated datasets.

| Model | ChatGPT Generations | | | | | | | | Llama2-Chat 70B Generations | | | | | | | |
|---|---|---|---|---|---|---|---|---|---|---|---|---|---|---|---|---|
| | ent | rel | con | inv | subj | unv | OA | Bi. | ent | rel | con | inv | subj | unv | OA | Bi. |
| CGPT | 12.1 | 25.0 | 50.0 | 6.7 | 7.7 | 0.0 | 13.0 | 35.1 | 17.5 | 13.2 | 36.4 | 15.5 | 18.8 | 9.5 | 19.3 | 59.2 |
| R+CGPT | 19.5 | 12.5 | 20.6 | 3.3 | 28.6 | 11.9 | 17.2 | 49.7 | 25.2 | 16.7 | 18.8 | 25.7 | 14.9 | 3.2 | 21.0 | 60.0 |
| Ours | 35.4 | 20.0 | 46.2 | 10.0 | 75.0 | 30.0 | 40.1 | 69.1 | 56.8 | 31.3 | 23.7 | 39.6 | 70.6 | 46.4 | 46.1 | 80.0 |

Table 13: Fine-grained detection task (Precision). CGPT indicates ChatGPT. "OA" indicates overall accuracy and "Bi." indicates binary predictions accuracy.

| Model | ChatGPT Generations | | | | | | | | Llama2-Chat 70B | | | | | | | |
|---|---|---|---|---|---|---|---|---|---|---|---|---|---|---|---|---|
| | ent | rel | con | inv | subj | unv | OA | Bi. | ent | rel | con | inv | subj | unv | OA | Bi. |
| CGPT | 51.3 | 33.3 | 33.3 | 50.0 | 7.7 | 0.0 | 33.6 | 87.4 | 41.8 | 19.2 | 21.1 | 8.5 | 16.7 | 20.0 | 32.4 | 81.1 |
| R+CGPT | 49.9 | 41.7 | 33.3 | 14.3 | 55.6 | 22.4 | 41.7 | 93.1 | 50.9 | 45.8 | 33.3 | 19.6 | 22.2 | 8.8 | 41.0 | 92.5 |
| Ours | 55.2 | 33.3 | 75.0 | 50.0 | 66.7 | 42.9 | 55.5 | 90.0 | 55.5 | 38.5 | 33.3 | 30.6 | 63.7 | 40.6 | 46.8 | 81.2 |

Table 14: Fine-grained detection task (Recall). CGPT indicates ChatGPT. "OA" indicates overall accuracy and "Bi." indicates binary predictions accuracy.

## C.2 Benchmark Prompts

Table 15 shows the prompt used for baseline models when evaluating on the detection benchmark.

## C.3 Details of Human Evaluation on Model Outputs

We conduct a human evaluation on 50 detection results of the top two models, namely Fava, and Rt-ChatGPT. Our results are shown in Table 16. We anonymize the model results to avoid potential biases during evaluations. Annotators count how many of the detection and editing results are correct given extensive web searches.

## C.4 Manual Analysis on Generated Data for Fava training

We conduct human evaluations on 50 generated data to assess the automatic data creation quality. Prior work introduces entity-centric synthetic data creation Longpre et al. (2021) and often results in unrealistic edits that powerful LMs can easily identify which are synthetically perturbed. Therefore, we evaluate not only *validity* but also *quality*. In particular, we evaluate

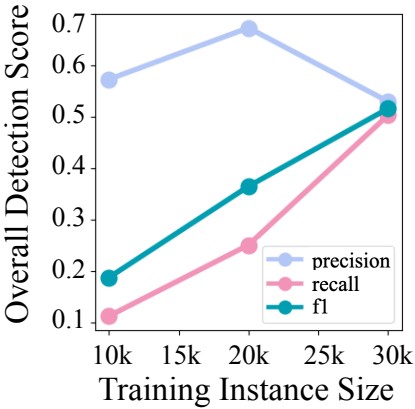

Figure 7: Training Size

a generated instance from two aspects: (1) *validity*—whether the model edits violate our annotation scheme; (2) *quality*—the inserted errors are feasible and realistic and notably different from provided few-shot samples. We ask human annotators to score each category for each passage either 0, 1, or 2 (higher is better).

Our analysis revealed that the data generated by our system with iterative insertion obtained an average score of 1.66 for validity assessment and 1.36 for quality assessment. Meanwhile, data generated by one-shot prompting scored an average of 1.1 for validity and 0.9 for quality. Our human evaluation reveals that our LM-generated edits are indeed of higher quality than the one-shot data creation, and provide realistic factual errors.

Table 17 shows examples of generated erroneous data from our synthetic data creation pipeline. We ask human annotators to score each category for each passage either 0, 1, or 2. These scores were defined as follows: a score of 0 indicated that less than half of the inserted errors aligned with the given criteria, a score of 1 signified that at least half of the errors met the established criteria, and a score of 2 indicated that all inserted errors met the criteria. As a comparison, we also generate training data with one-shot generation.

### C.5 More Ablations

**Retrieval ablations.** Table 18 shows the full results of the retrieval ablations. We enhance our retrieval process by strategically reordering the retrieval results. This involves rearranging the top 5 passages from our reranked Contriever documents and entity matching document such that the documents with the highest relevance are positioned closest to the text requiring verification. Reordering passages gives 1.7% improvements, further emphasizing the importance of retrieval and careful pipeline design.

**Analysis on data scaling.** We conduct ablations of different data scaling in FAVA. Figure 7 shows the scaling results.

```
Given a passage with factual errors, identify any <entity>, <relation>,
<contradictory>, <subjective>, <unverifiable> or <invented> errors in the passage and
add edits for <entity> and <relation> errors by inserting additional  or
<delete></delete> tags  to mark and delete. If there are no errors, return the passage
with no tags. Any changes to the original passage should be marked in <> tags. Below
are the error definitions followed by examples of what you need to follow.
Definitions:
1. entity errors (<entity>): a small part of a sentence, often an entity (e.g.,
location name), is incorrect (usually 1-3 words). Entity errors often involve
noun phrases or nouns.
2. relational error (<relation>): a sentence is partially incorrect as a small part
(usually 1 - 3 words). Relational errors often involve verbs and are often the
opposite of what it should be.
3. contradictory sentence error (<contradictory>): a sentence where the
entire sentence is contradicted by the given reference, meaning the sentence
can be proven false due to a contradiction with information in the passage.
4. invented info error (< invented >): these errors refer to entities that are
not known or do not exist. This does not include fictional characters in books or movies.
invented errors include phrases or sentences which have unknown entities or
misleading information.
5. subjective sentence (<subjective>): an entire sentence or phrase that is subjective
and cannot be verified, so it should not be included.
6. unverifiable sentence (<unverifiable>): a sentence where the whole sentence or
phrase is unlikely to be factually grounded although it can be true, and the sentence
cannot be confirmed nor denied using the reference given or internet search, it is
often something personal or private and hence cannot be confirmed.
Follow the given example exactly, your task is to create the edited completion
with error tags <>:

##
Passage: Marooned on Mars is a science fiction novel aimed at a younger audience.
It was written by Andy Weir and published by John C. Winston Co. in 1952, featuring
illustrations by Alex Schomburg. It ended up having a readership of older boys despite
efforts for it to be aimed at younger kids. The novel inspired the famous Broadway
musical "Stranded Stars," which won six Tony Awards. The novel tells a story of being
stranded on the Purple Planet. I wish the novel had more exciting and thrilling plot
twists.

Reference: Marooned on Mars is a juvenile science fiction novel written by American
writer Lester del Rey. It was published by John C. Winston Co. in 1952 with illustrations
by Alex Schomburg.

Edited: Marooned on Mars is a science fiction novel aimed at a younger audience.
It was written by <entity>Lester del Rey<delete>Andy Weir</delete></entity>
and published by John C. Winston Co. in 1952, featuring illustrations by Alex Schomburg.
<contradictory>It ended up having a readership of older boys despite efforts for it to be
aimed at younger kids .</contradictory>. <invented>The novel inspired the famous Broadway
musical "Stranded Stars," which won six Tony Awards.</invented> The novel tells a story
of being stranded on the <entity>Red<delete>Purple</delete></entity> Planet.
<subjective>I wish the novel had more exciting and thrilling plot twists.</subjective>
##

Now detect errors and include edits in the following passage like done in the example above.
Include error tags <> for ANYTHING YOU CHANGE IN THE ORIGINAL PASSAGE.

Passage: [PASSAGE_TO_VERIFY]

Reference: [REFERENCE]

Edited:
```

Table 15: Prompt used for baseline models for detection benchmark. We insert the retrieved context at the [Reference:] portion for retrieval-augmented baselines. For non-retrieval-augmented baselines, we do not insert [Reference:] component.

| Editor | avg. $|\mathbf{E}|$ | Detect (%) | Edit (%) |
|---|---|---|---|
| Rt+ChatGPT | 1.9 | 23.9 | 17.2 |
| FAVA (ours) | 2.4 | **55.7** | **63.5** |

Table 16: Human evaluation results. We show the average number of detected errors and the correctness (%) of the fine-grained types and edits.

| Original Passage | Automatic Data Generations for Training |
|---|---|
| "Solvalou," the hyper-entertainment machine known for its sit-down cabinet design and 3D shooting gameplay, has gained critical acclaim. Although it failed to find commercial success, the game recently made a comeback with a digital re-release for the Japanese Wii Virtual Console in 2009 as one of the first titles under the Virtual Console Arcade brand. Despite an announced home port for the 3DO Interactive Multiplayer in 1994, the release was ultimately canceled and never reached the market. | "Solvalou," the hyper-entertainment machine known for its sit-down cabinet design and 3D shooting gameplay, has gained critical acclaim. ¡invented> The game won the Shooting Star Award in 2020 for its innovative concept.</invented> Although it failed to find commercial success, the game recently made a comeback with a digital re-release for the Japanese Wii Virtual Console in 2009 as one of the first titles under the Virtual Console Arcade brand. Despite an announced home port for the 3DO Interactive <entity> <delete> Multiplayer</delete>  Singleplayer  </entity> in 1994, the release was ultimately canceled and never reached the market. <subjective> The game is a masterpiece that revolutionized the arcade industry and will always be remembered as a classic.</subjective> |
| The knapsack problem is a problem in combinatorial optimization where the goal is to determine the optimal selection of items to maximize the total value within a given weight constraint. | The knapsack problem is a problem in <entity> <delete> combinatorial optimization</delete>  sequential search </entity> where the goal is to determine the optimal selection of items to maximize the total <entity> <delete> value</delete>  weight </entity> within a given weight constraint. |

Table 17: Examples of generated data from automatic data creation.

| Settings | FActScore |
|---|---|
| Alpaca 13B (no edits) | 42.5 |
| FAVA (Top 1) | 44.2 +1.7 |
| FAVA (Top 5) | 47.0 +4.5 |
| FAVA (Top 5) + reranked | 47.7 +5.2 |
| + entity search | 50.1 +7.6 |
| + order | 51.8 +9.3 |

Table 18: FActScore editing results with different retrieval settings.

