# OpenReview forum: "Fine-grained Hallucination Detection and Editing for Language Models"
_colmweb.org/COLM/2024/Conference — COLM_

### Official Review · Reviewer_AbYD · 2024-04-30

**Rating:** 3
**Confidence:** 5
**Ethics Flag:** 1

**Summary:**

This paper introduces a taxonomy of hallucinations in large language models (LMs) and proposes a novel task of automatic fine-grained hallucination detection. The authors construct an evaluation benchmark, FavaBench, with over 1,000 human judgments on LM outputs.  The authors then train FAVA, a retrieval-augmented LM, to detect and correct fine-grained hallucinations. The results show that FAVA significantly outperforms ChatGPT and GPT-4 on fine-grained hallucination detection and improves the factuality of LM-generated text.

**Reasons To Accept:**

(1) An important task;
(2) An new method to create synthetic data for hallucination detection

**Reasons To Reject:**

(1) The authors make a very poor literature about LLM hallucination. Just list two highly related studies

-The dawn after the dark: An empirical study on factuality hallucination in large language models

Both papers are similar in many aspects: similar taxonomy for hallucination, detection baseline, and metrics.

-Halueval: A large-scale hallucination evaluation benchmark for large language models

Both papers are similar in many aspects: task, detection baseline, and result findings.

In addition, there are also several hallucination dection methods that are not compared by this work. Thus, I find that the literature is highly incomplete.

(2) The method to create synthetic data for hallucination detection, while a side effect is that LLM-generated errors may not be similar to the actual genrations of LLMs. That is, the detection method may only learn some shortcut patterns to detect hallucinations.

---

> ### Author Rebuttal · Authors · 2024-05-31
>
> We thank the reviewer for dedicating time to evaluate our paper.
> #Reasons to Reject 1: Related work
> Thank you for suggesting papers. We are happy to include those two papers in related work in our final version. However, we respectfully disagree with the reviewer’s assessment that both papers are “similar in many aspects: task, detection baseline, and result findings.”
> Below, we discuss the relationship between our work and suggested papers [1][2].
> [1] HaluEval: A Large-Scale Hallucination Evaluation Benchmark for Large Language Models.
> [2] The Dawn After the Dark: An Empirical Study on Factuality Hallucination in Large Language Models.
> Hallucination taxonomy: Note that [2] is unpublished and was posted on arXiv less than three months before the COLM deadline while our paper was in review. [2] introduces a taxonomy with 6 hallucination types, but our taxonomy is notably distinct, with 3 completely different types. While both studies identify "entity," "relation" types, these types have been used in prior work  (Pagnoni et al., 2021) and as we discussed in Section 3, our hallucinations extend from prior research. Although both taxonomies include "unverifiable" types, our approach distinguishes between "invented," "subjective," and "unverifiable" claims, while [2] groups these into one category.
> Detection baseline/metrics: [1] and [2]  are primarily within a binary detection framework, while our research introduces a novel approach by addressing fine-grained hallucinations and editing. Unlike binary detection, our system conducts evaluations at a span-level, identifying exact spans and categorizing them based on our hallucination taxonomy. Additionally, both binary/fine-grained detections are scored against human-annotated data in our study. Although [2] uses human annotators to validate some detections, our scoring is entirely based on human annotations. Also, our detection system incorporates editing which we evaluate using FActScore, a feature absent in [1][2].
> #Reasons to Reject 2: Quality of generated data
> Our synthetic data generation pipeline underwent carefully designed iterative generations to ensure similarity to hallucinations generated by LMs, complemented by continuous human evaluations. In Section 5.1 of our paper, we provide insights into the manual analysis techniques used to validate the generated data, with further elaboration in Appendix Section C4. We will move the details of human evaluations to the main page of our final version.

---

> ### Author Response · Authors · 2024-06-05
> **Follow-Up on Rebuttal Response**
>
> Dear Reviewer,
>
> Thank you for providing comments on our paper. We believe our rebuttal addresses your concerns, and we are happy to answer any further questions you may have.
>
> In our rebuttal, we provide a detailed discussion on the differences between this work and the two suggested papers, in terms of task setup, baselines, taxonomy, and the proposed method, and we believe our work and suggested papers are substantially different. We will add the suggested two papers to the Related Work of our final version.
>
> Please note that [1] was arXived on January 6, 2024, which is considered concurrent work according to the NLP/ML conferences' reviewing protocol e.g., [ACL review policies](https://www.aclweb.org/adminwiki/index.php/ACL_Policies_for_Review_and_Citation), and comparison of such work is not required
>
> > For comparison, papers (whether refereed or not) appearing less than 3 months before the submission deadline should be considered contemporaneous to the submission. This relieves authors from the obligation to make detailed comparisons that require additional experimentation and/or in-depth analysis
>
> [1] Junyi Li et al., The Dawn After the Dark: An Empirical Study on Factuality Hallucination in Large Language Models. Arxiv 2024.
>
>
>
> We also addressed your concern regarding the quality of synthetically generated data in our rebuttal, and we will move the details of our human evaluations in Appendix to the main pages in the final version.
>
> Thank you once again for your time and consideration. Please let us know if you have any follow-up concerns or questions.

---

### Official Review · Reviewer_Lc32 · 2024-05-11

**Rating:** 7
**Confidence:** 5
**Ethics Flag:** 1

**Summary:**

This paper proposes a new taxonomy for hallucinations and introduces a new fine-grained hallucination detection and editing task based on it. To evaluate this task, the authors provided FAVABENCH, a manually annotated benchmark. Additionally, instead of using existing LLMs for hallucination detection and editing, the authors also fine-tuned a retrieval-augmented LM called FAVA, which outperformed existing LLMs in hallucination detection and editing. The training of the FAVA model was conducted through synthetic data generation, which also follows the proposed taxonomy.

The paper is well-written, and the included figures significantly aid understanding. The topic of LLM hallucinations is also timely and appropriate, and the models and datasets provided by the authors would be beneficial to the community. However, apart from providing a taxonomy, there is no notable technological impact.

**Questions To Authors:**

- If Task 1, Fine-grained hallucination detection, involves binary classification for each error type, it's possible that a sentence might contain multiple error types. Then, for Task 2, Hallucination editing, how is editing performed according to each error type? Are there issues such as overlapping span boundaries in each edition?
- In fine-grained hallucination detection, does the process involve separate binary classifications for each error type, or can the classification of one error type influence another (e.g., multi-label classification)?
- Could you provide more detailed information about the third item on the list of source prompts for FAVABENCH, which states, "50 instructions requiring more reasoning and knowledge by the authors"?

**Reasons To Accept:**

- The paper is clearly well-written.
- The paper provides models and datasets (benchmarks).
- Well-designed experiments and their results demonstrate the usefulness of the proposed method.

**Reasons To Reject:**

I have some concerns about the appropriateness of the taxonomy proposed by the authors:

1. Entity and Relation errors: wouldn't it be possible that relation errors could also be resolved by replacing entities? Additionally, according to Figure 3, it seems that relation errors make up a very small percentage. Do relation errors introduce any unique impacts?
2. Ambiguity between Invented and Unverifiable: The difference between 'invented' and 'unverifiable' seems ambiguous. In Table 1's examples, 'invented' could also occur due to a lack of evidence. The authors state, "Invented are the hallucinations where we can verify that some core entities or subjects of the sentences don’t exist," but how can we be sure that these aren't issues with the retrieval stage?
- Regarding 2, could you provide the agreement scores among human annotators specifically for 'invented' and 'unverifiable' errors? It appears that the currently submitted agreement scores do not specify this for each specific type.

---

> ### Author Rebuttal · Authors · 2024-05-31
>
> We thank the reviewer for acknowledging our experiment design, the timeliness of this topic, and our writing and organization.
> Reason to Reject 1: Relation errors - This classification is derived from prior research, "Understanding Factuality in Abstractive Summarization with FRANK: A Benchmark for Factuality Metrics." Although making up a small percentage of hallucinations, we included this type to ensure our taxonomy's comprehensiveness. Our goal was to create a novel and comprehensive taxonomy for all hallucination types in information-seeking scenarios, justifying the inclusion of relation errors.
> Reason to Reject 2: Unverifiable vs Invented - We recognize the confusion between unverifiable and invented types. Invented types involve entirely fabricated entities, while unverifiable types lack supporting references. Our motivation for including both was to ensure thoroughness. Although some invented types could be unverifiable, not all unverifiable statements are invented. In our final version, we will clearly distinguish these categories and explain their inclusion.
> Responses to questions:
> Fine-grained edits: A sentence may include multiple hallucination types, but their spans do not overlap, preventing editing confusion. In the editing phase, deletion and insertion are prevalent on "entity" or "relation" types to correct factual accuracy. "Unverifiable" or "subjective" types are considered at the sentence level and highlighted for transparency, as they lack grounding in truth. "Contradictory" and "invented" types are also sentence level and typically deleted entirely to maintain text integrity.
> Error type classification: In fine-grained detection, each error type is assessed independently, with the classification of one type having no impact on the classification of another. When evaluating for fine-grained hallucinations, our analysis focuses on the presence of each hallucination type within each sentence in comparison to our human annotation detections. This approach ensures a thorough examination of the text, allowing for nuanced detection and classification of errors.
> Knowledge intensive dataset: The "50 instructions by the authors" dataset was created to enrich the annotation dataset with complex prompts. These instructions were meticulously crafted to enhance the dataset's diversity/comprehensiveness. While our dataset included prompts from sources like WebNLG, OpenAssistant, and NoRobot, we added these to broaden the scope across more scenarios.

---

> > ### Comment · Reviewer_Lc32 · 2024-06-03
> >
> > Thank you for your reply but I think you missed one of my questions.
> >
> > In Reasons To Reject:
> >
> > "Regarding 2, could you provide the agreement scores among human annotators specifically for 'invented' and 'unverifiable' errors? It appears that the currently submitted agreement scores do not specify this for each specific type."
> >
> > Would you be able to provide the scores?

---

> > > ### Author Response · Authors · 2024-06-05
> > >
> > > We apologize for overlooking that question in our response. We have calculated the cohen-kappa agreement scores for both types, getting 0.356 for 'unverifiable' and 0.636 for 'invented' across all our annotations. We will include a detailed breakdown of the agreement scores for each hallucination type in our camera-ready version.

---

> > > > ### Comment · Reviewer_Lc32 · 2024-06-06
> > > >
> > > > The Cohen-Kappa agreement scores for "unverifiable" are quite low, which could negatively affect the reasonability of the taxonomy. Have you analyzed the reasons for these low scores?

---

### Official Review · Reviewer_tF2X · 2024-05-11

**Rating:** 7
**Confidence:** 3
**Ethics Flag:** 1

**Summary:**

The authors introduce a comprehensive taxonomy that categorizes hallucinations into six detailed types and develop a novel fine-grained hallucination detection task. They also present FAVABENCH, an innovative benchmark for evaluating LM outputs with approximately 1,000 fine-grained human annotations. Leveraging this benchmark, the authors develop FAVA, a retrieval-augmented LM trained on synthetic data designed to detect and correct hallucinations at the span level. The evaluations demonstrate that FAVA significantly outperforms existing systems in both fine-grained and binary hallucination detection, thereby providing a robust solution for enhancing the factuality and reliability of LM-generated text.

**Reasons To Accept:**

1. The introduction of a comprehensive taxonomy that classifies hallucinations into six distinct categories based on their characteristics and the sources from which they need verification is a significant novelty. This classification allows for a more nuanced understanding and detection of errors than the traditional binary models of hallucination detection.

2. The creation of FAVA, a retrieval-augmented LM that operates at the span-level for both detection and correction of hallucinations, further underscores the innovative nature of the work.

3. The introduction of FAVABENCH as a new standard for evaluating language models could become a pivotal resource for future studies, encouraging more detailed and precise improvements in model outputs.

4. The paper is well-organized. Each section logically builds upon the previous, with clear explanations that make the sophisticated content accessible to readers with a background in the field.

**Reasons To Reject:**

1. LLMs are generally trained for broad applications, yet the proposed task demands detailed, fine-grained operations. Therefore, it is plausible that a smaller, task-specific LLM might outperform its more generic counterparts. However, the assertion that smaller LLMs exhibit remarkable performance over models like GPT and Llama is not definitive. If these larger models were trained using the same specialized dataset, one might anticipate even more significant improvements than the smaller, task-tailored models.

2. The authors dedicate substantial space to discussing the proposed task and datasets; however, the description of the experimental settings, especially in Section 7, lacks clarity. The absence of detailed experimental setup information for Table 4 complicates the distinction between the generators used in Tables 3 and 4. This ambiguity significantly delays comprehension of the comparative analysis. I recommend that the authors refine the description of the experimental setup.

---

> ### Author Rebuttal · Authors · 2024-05-31
>
> We’d like to thank the reviewer for their positive remarks on our taxonomy, the FAVA model, and our innovative benchmark, FAVABENCH. Additionally, we value their recognition of our writing clarity and organizational approach.
> # Reasons to Reject 1: Current Baselines
> We recognize that comparing task-specific models to general-purpose ones may yield anticipated results. Our primary objective in including baselines of GPT and Llama models was to underscore the necessity of a fine-tuned model for detecting fine-grained hallucinations. General-purpose models, despite the widespread recognition of their capabilities, struggle in this regard. By incorporating FActScore as a binary detection baseline, we aimed to contrast our approach with a specialized baseline tailored to error detection. Furthermore, due to the novelty of fine-grained detection or editing tasks, there is a scarcity of open-source models specialized for this purpose. Because of this, we were unable to identify comparable models trained on fine grained detection tasks for comparison. Therefore, we opted for carefully designed prompts and general-purpose models such as ChatGPT to fulfill this need.
> # Reasons to Reject 2: Lack of details in experiment section
> We acknowledge the reviewer's concern regarding the clarity of the experimental settings. While we prioritize providing context by dedicating lots of space to discussing tasks and datasets, we recognize that this may detract from the section on experimental setup details. In our camera-ready version, we will utilize the additional space to enhance the clarity of our experiment explanations and analyses, ensuring that the details are more comprehensible.

---

### Official Review · Reviewer_eZfJ · 2024-05-14

**Rating:** 5
**Confidence:** 4
**Ethics Flag:** 1

**Summary:**

This paper describes an approach to detect and edit fine-grained hallucinations. Several fine-grained hallucination types are defined. A strategy is proposed to generate hallucinations by prompting LLMs. A model is trained on generated data containing references-hallucination-correct triples. The model is asked to detect hallucinations and to edit hallucinations to produce the correct text.
Experiments are done on a set of cases. The method is compare to the ones based on LLMs, and shown to outperform the latter.

**Questions To Authors:**

Fig. 2: the tests used in the figure to determine types of hallucination are difficult to follow. For example, from the question "Does the information contradict with evidence?", there are two yes outputs, and from the "Is it verifiable based off web knowledge?" test, there is only one no output. The criteria to determine the types are confusing.

While it is interesting to include more (fine-grained) types of hallucination, it is unclear if "subjective" should be included as hallucination. The risk of considering it as hallucination is that many utterances in everyday dialogues (e.g. "I think ...") would be considered hallucinations. This does not make this type of hallucination very interesting.  What is the reason to include this type of hallucination?

On the other hand some other types of hallucination may require broad knowledge on the topic (invented, unverifiable). Looking at the approach, it seems that only 5 reference documents are included to cover the topic. This is not sufficient to verify if a statement is invented or verifiable. To detect these types of hallucination, more references (or larger models) should be considered. Therefore, the experiments done in the paper may not reflect the difficulty to detect these types of hallucination. Have you looked at the coverage of the 5 references on the topic? Are there cases where the statement is unverifiable with the 5 reference texts, but is verifiable with more reference texts?

The type "invented" is interesting. In addition to require broad knowledge to detect this type of hallucination, the generated data should also try to generate statement that are close to the topic. Looking at the examples, it seems that the "invented" hallucinations are randomly generated. This makes it artificially easy to detect, by checking if the topic of the statement is related to the remaining of the text. Have you tested different ways to generate invented hallucinations?

Table 4: I guess the numbers are FActScore. Please specify.

For hallucination editing, I believe the Human evaluation results are more meaningful than FActScore. It mat be appropriate to move Human evaluation results to the main body of the paper (instead of appendix).

The experiments only compare with simple LLM-based methods. There are more methods proposed in the literature. The paper mentions several ones in the related work. Another strongly related work is https://arxiv.org/pdf/2401.03205, which also detects hallucination cases and uses different methods to mitigate hallucinations, such as fine-tuning, different promptings, RLHF, etc. It is appropriate to compare with these approaches.

**Reasons To Accept:**

The categorization of hallucination is slightly different from the existing ones. This leads to more fine-grained types of hallucination.
The way to generate hallucination cases is interesting. LLMs are instructed to insert hallucinations from a correct text. This allows the authors to create a quite large set of hallucination cases (35k).
The use of a specifically trained model for hallucination detection and editing is reasonable.

**Reasons To Reject:**

Although the authors claim the types of fine-grained hallucination types are new, they are not very different from what others have proposed. The main categories are similar.
The details for training the model based on the generated data are missing.
The baselines considered in the paper are simple. In the literature, other approaches have been developed (e.g. checking factuality, fine-tuning a model. etc.). These models should also be included a baselines.

---

> ### Author Rebuttal · Authors · 2024-05-31
>
> We thank the reviewer for positive comments on our taxonomy and data generation process.
> Reason to Reject 1: Taxonomy novelty - While some types may be similar to prior work, our taxonomy as a whole is notably distinct from other work. In section 3.1, we acknowledge that our taxonomy is inspired by prior work but includes new types, which supports its novelty. Also, there isn't any recent work which includes "contradictory", "invented", or "subjective" types. Also, we ensure comprehensive coverage of significant hallucination types.
> Reason to Reject 2: Training details - Due to strict page limits, we included minimal training details in section 5.2 and more in appendix section B. We will move some details from the appendix to the main pages in the final version.
> Reasons to Reject 3: Baselines - Regarding baselines, we use FActScore, cited in related work, as a baseline for binary detection, shown in Table 3. Due to the novelty of fine-grained detection, there are no baselines from prior work; most prior studies focus on binary detection, making it challenging to apply their methods to fine-grained setups.
> Thank you for suggesting improvements to the presentation. We will implement these in our final version. Below, we address questions:
> Comparison with baselines, such as [1]: Comparing with [1] is difficult as the paper does not provide open-sourced models. Note that [1] was posted on arXiv within 3 months of the COLM deadline and is unpublished, considered concurrent work in many NLP/ML venues, hence not included in our work.
> Invented error: The prompt for this type of hallucination was carefully constructed after multiple iterations and manual analysis. We detail our manual analysis techniques and results in appendix section C4.
> Subjective statements: Our paper focuses on information-seeking contexts, classifying hallucinations as factually incorrect or unverifiable statements. Subjective statements are inherently not grounded in factual data and hence hallucinations in our context. We included this type to ensure the completeness of our taxonomy.
> Retrieval context size: We limited reference documents to five due to context size constraints but conducted ablations to ensure quality, detailed in section 7.2. For fine-grained detection, we compared our model against human-annotated data. Despite annotators having access to more resources, our model performed strongly, suggesting five documents were sufficient. We will discuss this in the final paper.

---

> ### Author Response · Authors · 2024-06-05
> **Follow-Up on Rebuttal Response**
>
> Dear Reviewer,
>
> Thank you for providing comments on our paper. Hopefully, our rebuttal addresses your concerns, and we are happy to answer any further questions you may have.
>
> In summary:
>
> - Our taxonomy is inspired by, yet substantially different from, prior studies. For example, "invented" or "subjective" are often grouped as "unverifiable." Our careful human annotations on 1k LM responses in multiple domains show that most hallucinations can be classified into one of these categories.
> - Regarding the baselines, we indeed compare our method with other state-of-the-art methods discussed in related work, such as FActScore, which first extracts a set of atomic statements and validate each of them using retrieved passages, and is widely used for hallucination detaction. Please note that the suggested paper [1] was arXived on January 6, 2024, which is considered concurrent work according to the NLP/ML conferences' reviewing protocol (e.g., ACL review policies [2]), and comparison of such work is not required.
> - We will include more details of our training in the final version of the paper.
>
>
> [1] Junyi Li et al., The Dawn After the Dark: An Empirical Study on Factuality Hallucination in Large Language Models. Arxiv 2024.
>
> [2] ACL Policies for Review and Citation https://www.aclweb.org/adminwiki/index.php/ACL_Policies_for_Review_and_Citation
> > For comparison, papers (whether refereed or not) appearing less than 3 months before the submission deadline should be considered contemporaneous to the submission. This relieves authors from the obligation to make detailed comparisons that require additional experimentation and/or in-depth analysis

---

> > ### Comment · Reviewer_eZfJ · 2024-06-07
> > **reaction to author's rebuttal**
> >
> > It is fine to not include a comparison with the related woik [1] according to ACL policy.
> > How about comparing with other methods that check for factuality? This is very similar to your verification using retrieved documents.
> > The new categories are slightly different from the existing ones. The newly introduced categories have the problems mentioned in the review. In the rebuttal, the authors said that "subjective" depend on factuality. Then what difference can be made between "subjective" and "unverifiable"? The justification of the new categories seems weak and they are not well defined.
> > Despite the explanation in the rebuttal, the main problems mentioned in the review are still there and not solved.

---

### Decision · Program_Chairs · 2024-07-10

**Decision:**

Accept

**Comment:**

This paper introduces a new benchmark, FavaBench, for detecting fine-grained hallucinations in large language model outputs. The authors propose a novel taxonomy for hallucinations and develop a retrieval-augmented model (FAVA) to detect and correct these errors. Evaluation results indicate that FAVA significantly outperforms several frontier LLMs in hallucination detection and correction tasks. A reviewer raised some criticisms regarding its literature review and comparisons with concurrent work -- I personally really liked this work, and I believe it significantly improves our ability to understand and mitigate the impact of hallucinations in LLM-generated texts.